# Delving into the Cyclic Mechanism in Semi-supervised Video Object Segmentation

**Yuxi Li**[*]
Shanghai Jiao Tong University
Shanghai, China
lyxok1@sjtu.edu.cn

**Ning Xu**[*]
Adobe Research
San Jose, CA
nxu@adobe.com

**Jinlong Peng**[*]
Tencent Youtu Lab
Shanghai, China
jeromepeng@tencent.com

**John See**
Multimedia University
Selangor, Malaysia
johnsee@mmu.edu.my

**Weiyao Lin**[†]
Shanghai Jiao Tong University
Shanghai, China
wylin@sjtu.edu.cn

## Abstract

In this paper, we address several inadequacies of current video object segmentation pipelines. Firstly, a cyclic mechanism is incorporated to the standard semi-supervised process to produce more robust representations. By relying on the accurate reference mask in the starting frame, we show that the error propagation problem can be mitigated. Next, we introduce a simple gradient correction module, which extends the offline pipeline to an online method while maintaining the efficiency of the former. Finally we develop cycle effective receptive field (cycle-ERF) based on gradient correction to provide a new perspective into analyzing object-specific regions of interests. We conduct comprehensive experiments on challenging benchmarks of DAVIS17 and Youtube-VOS, demonstrating that the cyclic mechanism is beneficial to segmentation quality.

## 1 Introduction

Video object segmentation (VOS) is garnering more attention in recent years due to its widespread application in the area of video editing and analysis. Among all the VOS scenarios, semi-supervised video object segmentation is the most practical and widely researched. Specifically, a mask is provided in the first frame indicating the location and boundary of the objects, and the algorithm should accurately segment the same objects from the background in subsequent frames.

A natural solution toward the problem is to process videos in sequential order; this exploits the information from previous frames and guides the segmentation process in the current frame. In most practical scenarios, the video is processed in an online manner where only previous knowledge is available. Due to this reason, most state-of-the-art pipelines [1, 2, 3, 4, 5, 6, 7, 8, 9] follow a sequential order for segmentation in both training and inference stages. Ideally, if the masks predicted for intermediate frames are sufficiently accurate, they can provide more helpful object-specific features and position prior to segmentation. Besides, the existence of prediction errors at intermediate frames can be problematic — these masks can mislead the segmentation procedure in future frames. Figure 1 illustrates an example of such error propagation risk in sequential video object segmentation pipelines. As the algorithm is misled by another camel of similar appearance in the background, the segmented

---

[*]equal contribution

[†]Correspondance author

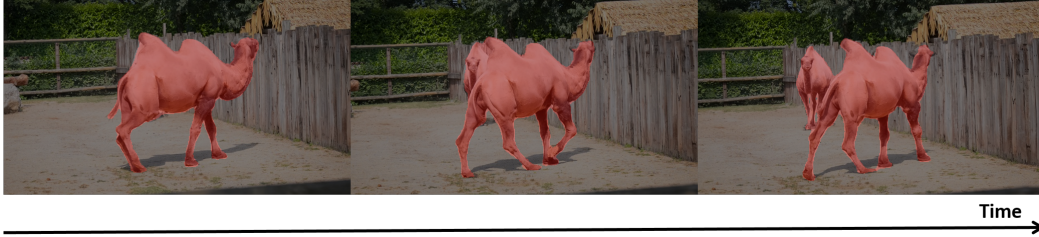

Figure 1: An example of error propagation risk during the inference time.

background camel will serve as erroneous guidance to future frames. Consequently the algorithm will gradually focus on both the foreground and background objects in upcoming new frames.

Based on these observations, in this paper, we propose to train and apply a segmentation network in cyclical fashion. In contrast to the predicted reference masks, the initial reference mask provided in the starting frame is always perfectly accurate and reliable (under semi-supervised mode). This inspires us to explicitly bridge the relationship between the initial reference mask and objective frame by taking the first reference mask as a measurement of prediction. This way, we can further refine the intermediate mask prediction and guide the network to learn more robust feature representation of cross frame correspondence, which is less prone to background distractors.

In this work, we apply a forward-backward data flow to form a cyclical structure, and train our segmentation network at both the objective frame and starting frame to help our model learn more robust correspondence relationship between predictions and the initial reference mask. Further to this, at the inference stage, we design a gradient correction module to selectively refine the predicted mask based on the gradient backward from the starting frame. In this way, we are able to naturally extend the offline trained model to an online scheme with marginal increase in time latency. Furthermore, we train our model under such cyclic consistency constraint without additional annotated data from other tasks. The trained models are evaluated in both online and offline schemes on common object segmentation benchmarks: DAVIS17 [10] and Youtube-VOS [11], in which we achieve results that are competitive to other state-of-the-art methods while keeping the efficiency on par with most offline approaches.

Additionally, inspired by the process of gradient correction, we develop a new receptive field visualization method called cycle effective receptive field (cycle-ERF), which gradually updates an empty objective mask to show the strong response area w.r.t. the reference mask. In our experiments, we utilize the cycle-ERF to analyze how the cyclic training scheme affects the support regions of objects. This visualization method provides a fresh perspective for analyzing how the segmentation network extracts regions of interests from guidance masks.

In a nutshell, the contribution of this paper can be summarized as follows:

- We incorporate cycle consistency into the training process of a semi-supervised video object segmentation network to mitigate the error propagation problem. We achieved competitive results on mainstream benchmarks.

- We design a gradient correction module to extend the offline segmentation network to an online approach, which boosts the model performance with marginal increase in computation cost.

- We develop cycle-ERF, a new visualization method to analyze the important regions for object mask prediction which offers explainability on the impact of cyclic training.

## 2 Related works

### 2.1 Semi-supervised video object segmentation

Semi-supervised video object segmentation has been widely researched in recent years with the rapid development of deep learning techniques. Depending on the presence of a learning process during inference stage, the segmentation algorithms can be generally divided into *online* methods and *offline*

methods. OVOS [2] is the first online approach to exploit deep learning for the VOS problem, where a multi-stage training strategy is design to gradually shrink the focus of network from general objects to the one in reference masks. Subsequently, OnAVOS [9] improved the online learning process with an adaptive mechanism. MaskTrack [6] introduced extra static image data with mask annotation and employed data synthesized through affine transformation, to fine-tune the network before inference. All of these online methods require explicit parameter updating during inference. Although high performance can be achieved, these methods are usually time-consuming with a real-time FPS of less than 1, rendering them unfeasible for practical deployment.

On the other hand, there are a number of offline methods that are deliberately designed to learn generalized correspondence feature and they do not require a online learning process during inference time. RGMP [1] designed an hourglass structure with skip connections to predict the objective mask based on the current frame and previous information. S2S [11] proposed to model video object segmentation as a sequence-to-sequence problem and proceeds to exploit a temporal modeling module to enhance the temporal coherency of mask propagation. Other works like [8, 4] resorted to using state-of-the-art instance segmentation or tracking pipeline [12, 13] while attempting to design matching strategies to associate the mask over time. A few recent methods FEELVOS [7] and AGSS-VOS [3] mainly exploited the guidance from the initial reference and the last previous frame to enhance the segmentation accuracy with deliberately designed feature matching scheme or attention mechanism. STM [5] further optimized the feature matching process with external feature memory and an attention-based matching strategy. Compared with online methods, these offline approaches are more efficient. However, to learn more general and robust feature correspondence, these data-hungry methods may require backbones pretrained on extra data with mask annotations from other tasks such as instance segmentation [14] or saliency detection [15]. Without these auxiliary help, the methods might well be disrupted by distractions from similar objects in the video, which then propagates erroneous mask information to future frames.

## 2.2 Cycle consistency

Cycle consistency is widely researched in unsupervised and semi-supervised representation learning, where a transformation and its inverse operation are applied sequentially on input data, the consistency requires that the output representation should be close to the original input data in feature space. With this property, cycle consistency can be applied to different types of correspondence-related tasks. [16] combined patch-wise consistency with a weak tracker to construct a forward-backward data loop and this guides the network to learn representative feature across different time spans. [17] exploited the cycle consistency in unsupervised optical flow estimation by designing a bidirectional consensus loss during training. On the other hand Cycle-GAN [18] and Recycle-GAN [19] and other popular examples of how cyclic training can be utilized to learn non-trivial cross-domain mapping, yielding reliable image-to-image transformation across different domains.

Our method with cyclic mechanism is different from the works mentioned above in two main aspects. First, we incorporate cycle consistency with network training in a fully supervised manner, without requiring large amounts of unlabeled data as [16]. Second, our cyclic structure is not only applicable during training, but also useful in the inference stage.

## 3 Methods

### 3.1 Problem formulation

Given a video of length $T$, $X_t$ is the $t$-th frame ($t \in [1, T]$) in temporal sequential order, and $Y_t$ is its corresponding annotation mask. $\mathcal{S}_\theta$ is an object segmentation network parameterized by learnable weights $\theta$. In terms of the sequential processing order of the video, the segmentation network should achieve the function as in Equation (1) below:

$$\widehat{Y}_t = \mathcal{S}_\theta \left( \mathcal{X}_{t-1}, \mathcal{Y}_{t-1}, X_t \right) \quad t \in [2, T] \tag{1}$$

where $\widehat{Y}_t$ denotes the predicted object mask at $t$-th frame. $\mathcal{X}_{t-1} \subset \{X_i | i \in [1, t-1]\}$ is the reference frame set, which is a subset of all frames appearing before objective frame $X_t$. Similarly, $\mathcal{Y}_{t-1}$ is a set containing reference object masks corresponding to the reference frames in $\mathcal{X}_{t-1}$. However, in the semi-supervised setting, only the initial reference mask at the first frame is available. Therefore, in

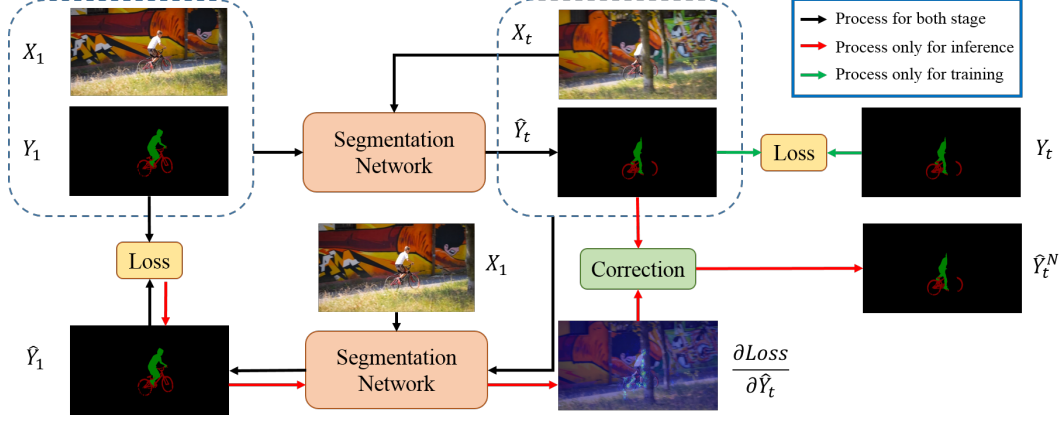

Figure 2: Overview of the proposed cyclic mechanism in both training and inference stages of the segmentation network. For simplicity, we take the situation where $\mathcal{X}_{t-1} = \{X_1\}$, $\mathcal{Y}_{t-1} = \{Y_1\}$, $\widehat{\mathcal{X}_t} = \{X_t\}$ and $\widehat{\mathcal{Y}_t} = \{\widehat{Y}_t\}$ as an example.

the inference stage, the corresponding predicted mask $\widehat{Y}_t$ is used as the approximation of the reference mask. Hence, we have $\mathcal{Y}_{t-1} \subset \{Y_1\} \bigcup \{\widehat{Y}_i | i \in [2, t-1]\}$.

## 3.2 Cycle consistency loss

For the sake of mitigating error propagation during training, we incorporate the cyclical process into the offline training process to explicitly bridge the relationship between the initial reference and predicted masks. To be specific, as illustrated in Figure 2, after obtaining the predicted output mask $\widehat{Y}_t$ at frame $t$, we construct a **cyclic reference set** for frames and mask set, respectively denoted as $\widehat{\mathcal{X}_t} \subset \{X_i | i \in [2, t]\}$, $\widehat{\mathcal{Y}_t} \subset \{\widehat{Y}_i | i \in [2, t]\}$.

With the cyclic reference set, we can obtain the prediction for the initial reference mask in the same manner as sequential processing:

$$\widehat{Y}_1 = \mathcal{S}_\theta \left( \widehat{\mathcal{X}_t}, \widehat{\mathcal{Y}_t}, X_1 \right) \tag{2}$$

Consequently, we apply mask reconstruction loss (in Equation 3) during supervision, optimizing on both the output $t$-th frame and the backward prediction $\widehat{Y}_1$.

$$\mathcal{L}_{cycle,t} = \mathcal{L}(\widehat{Y}_t, Y_t) + \mathcal{L}(\widehat{Y}_1, Y_1) \tag{3}$$

In implementation, we utilize the combination of cross-entropy loss and mask IOU loss as supervision at both sides of the cyclic loop, which can be formulated as,

$$\mathcal{L}(\widehat{Y}_t, Y_t) = \frac{1}{|\Omega|} \sum_{u \in \Omega} \left( (1 - Y_{t,u}) \log(1 - \widehat{Y}_{t,u}) + Y_{t,u} \log(\widehat{Y}_{t,u}) \right) - \gamma \frac{\sum_{u \in \Omega} \min(\widehat{Y}_{t,u}, Y_{t,u})}{\sum_{u \in \Omega} \max(\widehat{Y}_{t,u}, Y_{t,u})} \tag{4}$$

where $\Omega$ denotes the set of all pixel coordinates in the mask while $Y_{t,u}$ and $\widehat{Y}_{t,u}$ are the normalized pixel values at coordinate $u$ of the masks, $\gamma$ is a hyperparameter to balance between the two loss components. It should also be noted that the cyclic mechanism in Figure 2 indirectly applies data augmentation on the training data by reversing the input clips in temporal order, helping the segmentation network to learn more general feature correspondences.

## 3.3 Gradient correction

After training with the cyclic loss as Equation (3), we can directly apply the offline model in the inference stage. However, inspired by the cyclic structure in training process, we can take the accurate initial reference mask as a measurement to evaluate the segmentation quality of current frame and proceed to refine the output results based on the evaluation results. In this way, we can explicitly reduce the effect of error propagation during inference time.

To achieve this goal, we design a gradient correction block to update segmentation results iteratively as illustrated in Figure 2. Since only the initial mask $Y_1$ is available in inference stage, we apply the predicted mask $\widehat{Y}_t$ to infer the initial reference mask in the same manner as Equation (2), and we evaluate the segmentation quality of $\widehat{Y}_t$ with the loss function in Equation (4). Intuitively, a more accurate prediction mask $\widehat{Y}_t$ will result in a smaller reconstruction error for $Y_1$; therefore, the gradient descent method is adopted to refine the mask $\widehat{Y}_t$. To be specific, we start from an output mask $\widehat{Y}_t^0 = \widehat{Y}_t$, and then update the mask for $N$ iterations:

$$\widehat{Y}_t^{l+1} = \widehat{Y}_t^l - \alpha \frac{\partial \mathcal{L}\left(\mathcal{S}_\theta\left(\{X_t\}, \{\widehat{Y}_t^l\}, X_1\right), Y_1\right)}{\partial \widehat{Y}_t^l} \tag{5}$$

where $\alpha$ is a predefined correction rate for mask update. After $N$ iterations, we take the output $\widehat{Y}_t^N$ as the final segmentation. With this iterative refinement, we naturally extend the offline model to an online inference algorithm. However, the gradient correction approach can be time-consuming since it requires multiple times of network forward-backward pass. Due to this reason, we only apply gradient correction once per $K$ frames to achieve good performance-runtime trade-off.

### 3.4 Cycle-ERF

The cyclic mechanism with gradient update in Equation (5) is not only helpful for the output mask refinement, but it also offers a new aspect of analyzing the region of interests of specific objects segmented by the pretrained network. In detail, we construct a reference set as $\mathcal{X}_l = \{X_l\}$ and $\mathcal{Y}_l = \{\mathbf{0}\}$ as the guidance, where $\mathbf{0}$ denotes an empty mask of the same size as $X_l$ but is filled with zeros. We take these references to predict objects at the $t$-th frame $\widehat{Y}_t$. To this end, we can obtain the prediction loss $\mathcal{L}(\widehat{Y}_t, Y_t)$. To minimize this loss, we conduct the gradient correction process as in Equation (5) to gradually update the empty mask for $M$ iterations. Finally, we take the ReLU function to preserve the positively activated areas of the objective mask as our final cycle-ERF representation.

$$\textbf{cycle-ERF}(Y_l) = ReLU\left(\widehat{Y}_l^M\right) \tag{6}$$

As we will show in our experiments, the cycle-ERF is capable of properly reflecting the support region of specific objects for the segmentation task. Through this analysis, the pretrained model can be shown to be particularly concentrated on certain objects in video.

## 4 Experiments

### 4.1 Experiment setup

**Datasets.** We train and evaluate our method on two widely used benchmarks for semi-supervised video object segmentation, DAVIS17 [10] and Youtube-VOS [11]. DAVIS17 contains 120 video sequences in total with at most 10 objects in a video. The dataset is split into 60 sequences for training, 30 for validation and the other 30 for test. The Youtube-VOS is larger in scale and contains more object categories. There are a total of 3,471 video sequences for training and 474 videos for validation in this dataset with at most 12 objects in a video. Following the training procedure in [5, 7], we construct a hybrid training set by mixing the data from two training sets. Furthermore, we also report the results of other methods with Youtube-VOS pretraining if available.

**Metrics.** For evaluation on DAVIS17 validation and test set, we adopt the metric following standard DAVIS evaluation protocol [10]. The Jaccard overlap $\mathcal{J}$ is adopted to evaluate the mean IOU between predicted and groundtruth masks. The contour F-score $\mathcal{F}$ computes the F-measurement in terms of the contour based precision and recall rate. The final score is obtained from the average value of $\mathcal{J}$ and $\mathcal{F}$. The evaluation on Youtube-VOS follows the same protocol except that the two metrics are computed on seen and unseen objects respectively and averaged together.

**Baseline.** We take the widely used Space Time Memory Network (STM) [5] as our baseline model due to its flexibility in adjusting the reference sets $\mathcal{X}_t$ and $\mathcal{Y}_t$. However, since the original STM model involves too much external data and results in unfair comparison, we retrain our implemented model with only the training data in DAVIS17 and Youtube-VOS in all experiments. In order to adapt

| | validation | | | | | |
|---|---|---|---|---|---|---|
| Method | Extra data | OL | $\mathcal{J}(\%)$ | $\mathcal{F}(\%)$ | $\mathcal{J}\&\mathcal{F}(\%)$ | FPS |
| RGMP [1] | ✓ | | 64.8 | 68.6 | 66.7 | 3.6 |
| DMM-Net [8] | ✓ | | 68.1 | 73.3 | 70.7 | - |
| AGSS-VOS [3] | ✓ | | 64.9 | 69.9 | 67.4 | 10 |
| FEELVOS [7] | ✓ | | 69.1 | 74.0 | 71.5 | 2 |
| Official STM [5] | ✓ | | **79.2** | **84.3** | **81.8** | 6.3 |
| OnAVOS [9] | ✓ | ✓ | 61.0 | 66.1 | 63.6 | 0.04 |
| PReMVOS [4] | ✓ | ✓ | 73.9 | 81.7 | 77.8 | 0.03 |
| STM-cycle (Ours) | | | 68.7 | 74.7 | 71.7 | **38** |
| STM-cycle+GC (Ours) | | ✓ | 69.3 | 75.3 | 72.3 | 9.3 |
| | test-dev | | | | | |
| Method | Extra data | OL | $\mathcal{J}(\%)$ | $\mathcal{F}(\%)$ | $\mathcal{J}\&\mathcal{F}(\%)$ | FPS |
| RVOS [23] | | | 48.0 | 52.6 | 50.3 | 22.7 |
| RGMP [1] | ✓ | | 51.3 | 54.4 | 52.8 | 2.4 |
| AGSS-VOS [3] | ✓ | | 54.8 | 59.7 | 57.2 | 10 |
| FEELVOS [7] | ✓ | | 55.2 | 60.5 | 57.8 | 1.8 |
| OnAVOS [9] | ✓ | ✓ | 53.4 | 59.6 | 56.9 | 0.03 |
| PReMVOS [4] | ✓ | ✓ | **67.5** | **75.7** | **71.6** | 0.02 |
| STM-cycle (Ours) | | | 55.1 | 60.5 | 57.8 | **31** |
| STM-cycle+GC (Ours) | | ✓ | 55.3 | 62 | 58.6 | 6.9 |

Table 1: Comparison with state-of-the-art method on DAVIS17 validation and test-dev set. "Extra data" indicates the method is pretrained with extra data with mask annotations. "OL" denotes online learning or update process. "GC" is short for gradient correction.

| Method | Extra data | OL | $\mathcal{J}_{\mathcal{S}}(\%)$ | $\mathcal{J}_{\mathcal{U}}(\%)$ | $\mathcal{F}_{\mathcal{S}}(\%)$ | $\mathcal{F}_{\mathcal{U}}(\%)$ | $\mathcal{G}(\%)$ | FPS |
|---|---|---|---|---|---|---|---|---|
| RVOS [23] | | | 63.6 | 45.5 | 67.2 | 51.0 | 56.8 | 24 |
| S2S [11] | | | 66.7 | 48.2 | 65.5 | 50.3 | 57.6 | 6 |
| RGMP [1] | ✓ | | 59.5 | - | 45.2 | - | 53.8 | 7 |
| DMM-Net [8] | ✓ | | 58.3 | 41.6 | 60.7 | 46.3 | 51.7 | 12 |
| AGSS-VOS [3] | ✓ | | 71.3 | 65.5 | 75.2 | 73.1 | 71.3 | 12.5 |
| Official STM [5] | ✓ | | **79.7** | **84.2** | 72.8 | **80.9** | **79.4** | 6.3 |
| S2S [11] | | ✓ | 71.0 | 55.5 | 70.0 | 61.2 | 64.4 | 0.06 |
| OSVOS [2] | ✓ | ✓ | 59.8 | 54.2 | 60.5 | 60.7 | 58.8 | - |
| MaskTrack [6] | ✓ | ✓ | 59.9 | 45.0 | 59.5 | 47.9 | 53.1 | 0.05 |
| OnAVOS [9] | ✓ | ✓ | 60.1 | 46.6 | 62.7 | 51.4 | 55.2 | 0.05 |
| DMM-Net [8] | ✓ | ✓ | 60.3 | 50.6 | 63.5 | 57.4 | 58.0 | - |
| STM-cycle(Ours) | | | 71.7 | 61.4 | 75.8 | 70.4 | 69.9 | **43** |
| STM-cycle+GC(Ours) | | ✓ | 72.2 | 62.8 | **76.3** | 71.9 | 70.8 | 13.8 |

Table 2: Comparison with state-of-the-art method on Youtube-VOS validation set. The subscript $\mathcal{S}$ and $\mathcal{U}$ denote the seen and unseen categories. $\mathcal{G}$ is the global mean. "-" indicates unavailable results.

to the time-consuming gradient correction process, we take the lightweight design by reducing the intermediate feature dimension, resizing the input to half of the original work and upsampling the output to original size by nearest interpolation. For ease of representation, we denote the one trained with cyclic scheme as "STM-cycle".

**Implementation details.** The training and inference procedures are deployed on an NVIDIA TITAN Xp GPU. Within an epoch, for each video sequence, we randomly sample 3 frames as the training samples; the frame with the smallest timestamp is regarded as the initial reference frame. Similar to [5], the maximum temporal interval of sampling increases by 5 every 20 training epochs. We set the hyperparameters as $\gamma = 1.0$, $N = 10$, $K = 5$, and $M = 50$. The Resnet50 [20] pretrained on ImageNet [21] is adopted as our backbone in baseline. The network is trained with a batch size of 4 for 240 epochs in total and is optimized by the Adam optimizer [22] of learning rate $10^{-5}$ and $\beta_1 = 0.9, \beta_2 = 0.999$. In both training and inference stages, the input frames are resized to the resolution of $240 \times 427$. The final output is upsampled to the original resolution by nearest interpolation. For simplicity, we directly use $X_t$ and $\widehat{Y_t}$ to construct the cyclic reference sets.

## 4.2 Main results

**DAVIS17.** The evaluation results on DAVIS17 validation and test-dev set are reported in Table 1. From this table, we observe that our model trained with cyclic loss outperforms most of the offline

| | $\mathcal{J}(\%)$ | $\mathcal{F}(\%)$ | $\mathcal{J}\&\mathcal{F}(\%)$ |
|---|---|---|---|
| baseline | 67.6 | 71.7 | 69.7 |
| + cyclic | 68.7 | 74.7 | 71.7 |
| + GC | 67.8 | 72.8 | 70.3 |
| + both | **69.3** | **75.3** | **72.3** |

Table 3: Ablation study on the effectiveness of different component. "GC" is short for gradient correction.

| $\mathcal{X}_{t-1}$ | $\mathcal{Y}_{t-1}$ | baseline | +cycle | $\Delta$ |
|---|---|---|---|---|
| $\{X_1\}$ | $\{Y_1\}$ | 65.2 | 67.6 | +2.4 |
| $\{X_{t-1}\}$ | $\{\widehat{Y}_{t-1}\}$ | 56.8 | 61.2 | **+4.4** |
| $\{X_1, X_{t-1}\}$ | $\{Y_1, \widehat{Y}_{t-1}\}$ | 67.3 | 69.2 | +1.9 |
| **MEM** | **MEM** | 69.7 | 71.7 | +2.0 |

Table 4: Experiments on improvement of $\mathcal{J}\&\mathcal{F}$ score with different reference set configuration.

| low-quality mask | +GC | $\mathcal{J}(\%)$ | $\mathcal{F}(\%)$ | $\mathcal{J}\&\mathcal{F}(\%)$ |
|---|---|---|---|---|
| baseline predict | | 65.9 | 72.2 | 69.1 |
| | ✓ | **66.9** | **73.3** | **70.1** |
| bounding box | | 63.0 | 66.0 | 64.5 |
| | ✓ | **68.7** | **74.9** | **71.8** |

Table 5: Comparison with low-quality reference mask on DAVIS17 validation set.

methods and even performs better than the method with online learning [9]. When combined with the online gradient correction process, our method further improves. In terms of the runtime speed, although gradient correction increases the computation cost, our method still runs at a speed comparable to other offline methods [3] due to our efficient implementation. Although there is still a performance gap between our approach and the state-of-the-art online learning method [4] and official STM [5], our method is far more efficient and it does not aggressively collect extra data from instance segmentation tasks as training samples. It should also be noticed that our cyclic scheme is not structure-specific and can be potentially complementary to a general video object segmentation framework, e.g. the offical STM model can get at most 1.9 $\mathcal{J}\&\mathcal{F}$ gain when combined with gradient correction. This indicates that our scheme can potentially boost the performance of a general segmentation pipeline. The combination of our work and other VOS pipelines will be left as our future study.

**Youtube-VOS.** The evaluation results on Youtube-VOS validation set are reported in Table 2. On this benchmark, our model also outperforms some offline methods and their online learning counterparts [11, 8]. It is also noticeable that compared to the performance on seen objects, the one on unseen objects has improved more using our gradient correction strategy. Our final pipeline, however, performs slightly worse than [3] on Youtube-VOS, but our model runs faster even when added with gradient correction. We think this could be due to the lesser average number of objects in this benchmark.

### 4.3 Ablation study

In this section, we conduct ablation studies to analyze the impact of different components in our method, with all the experiments performed on the DAVIS17 validation set.

**Effectiveness of each component.** We first demonstrate the effectiveness of cyclic training and gradient correction in Table 3, where the baseline method [5] is implemented and retrained by ourselves. From this table, both components are helpful in boosting the performance. In particular, the incorporated cycle mechanism improves the contour score $\mathcal{F}$ more than the overlap, signifying that the proposed scheme is likely to be more useful for fine-grained mask prediction.

**Improvement with different reference sets.** Due to the flexibility of our baseline method in configuring its reference sets during inference, we tested how our cyclic training strategy impacts the performance using different reference sets. We conduct the test under four types of configuration: (1) Only the initial reference mask and its frame are utilized for predicting other frames. (2) Only the prediction of the last frame $\widehat{Y}_{t-1}$ and the last frame are used. (3) Both the initial reference and last frame prediction are utilized, which is the most common configuration in other state-of-the-art works. (4) The external memory strategy (denoted as **MEM**) in [5] is used where the reference set is dynamically updated by appending new prediction and frames at a specific frequency of 5Hz. In the results reported in Table 4, we observe that the cyclic training is helpful under all configurations. It is also interesting to see that our scheme achieves the maximum improvement (+4.6 $\mathcal{J}\&\mathcal{F}$) with

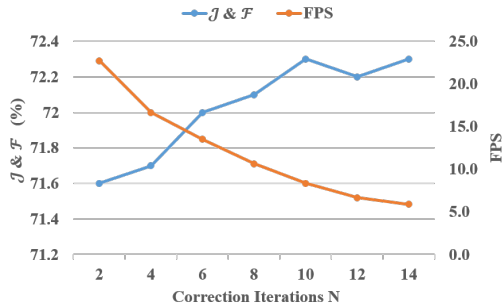
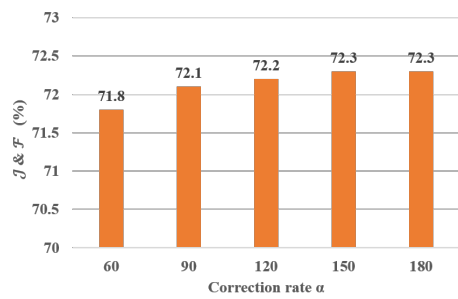

Figure 3: Performance-runtime trade-off with different iteration size $N$.

Figure 4: Performance with different correction rate.

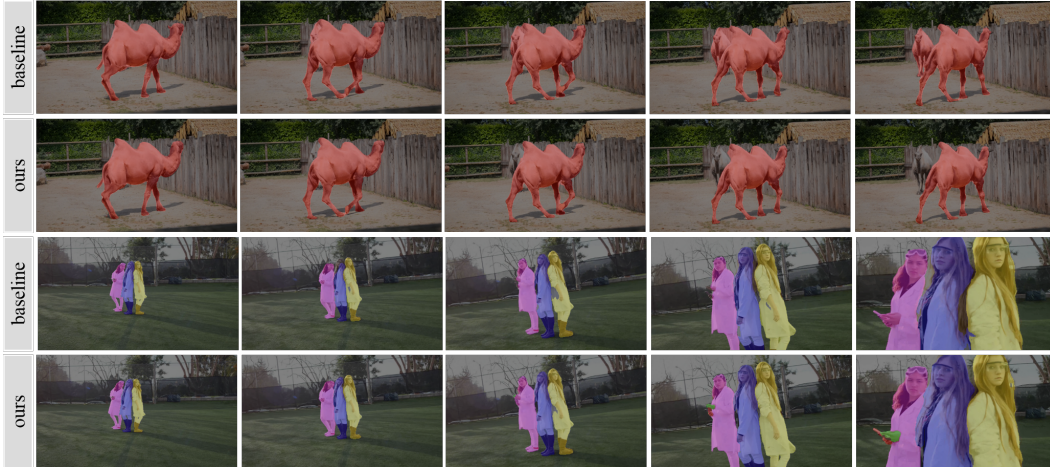

Figure 5: Qualitative results shows the improvement of cyclic training over the baseline.

the configuration $\mathcal{X}_{t-1} = \{X_{t-1}\}, \mathcal{Y}_{t-1} = \{\widehat{Y}_{t-1}\}$, since this case is the most vulnerable to error propagation.

**Sensitivity analysis.** Finally, we evaluate how the hyperparameters in our algorithm affect the final results. In Figure 3, we show the performance-runtime trade-off w.r.t. the correction iteration time $N$. We find that the $\mathcal{J}\&\mathcal{F}$ score saturates when $N$ approaches 10; above which, the score improvement is somewhat marginal but at the expense of decreasing efficiency. Therefore we take $N = 10$ as the empirically optimal iteration number for gradient correction. Additionally, we also analyze the impact of correction rate $\alpha$ as shown in Figure 4. We find the performance variation is not sensitive to the change of correction rate $\alpha$, reflecting that our update scheme is robust and can accommodate variations to this parameter well.

**Robustness to coarser reference.** Additionally, We further investigate how gradient correction process mitigate the effect of low-quality reference masks. To do this, our model is running with the **MEM** strategy as [5] by dynamically appending a predicted mask and its frame into the reference set. However, in this case, the predicted masks to be appended are manually replaced by a low-quality version. In our experiments, we take two adjustment schemes. (1) We replace the predicted mask $\widehat{Y}_t$ from baseline model on the same frame. This scheme is denoted as "baseline predict" (2) We replace the predicted mask $\widehat{Y}_t$ with a coarse level groundtruth mask where all pixels in the bounding box of objects are set to be 1. This scheme is denoted as "bounding box". For each scheme, we conduct another experiment with gradient correction on replaced masks before appending to the memory as the control group. From Table 5, we see the gradient correction is helpful for both low-quality reference condition. Especially, the improvement is much more obvious under the case of "bounding box", this indicates that gradient correction is more helpful when the intermediate reference mask is coarser but properly covers the object area.

## 4.4 Qualitative results

**Segmentation results.** In Figure 5, we show some segmentation results using the STM model trained with and without our cycle scheme. From comparison on the first sequences, we observe that the

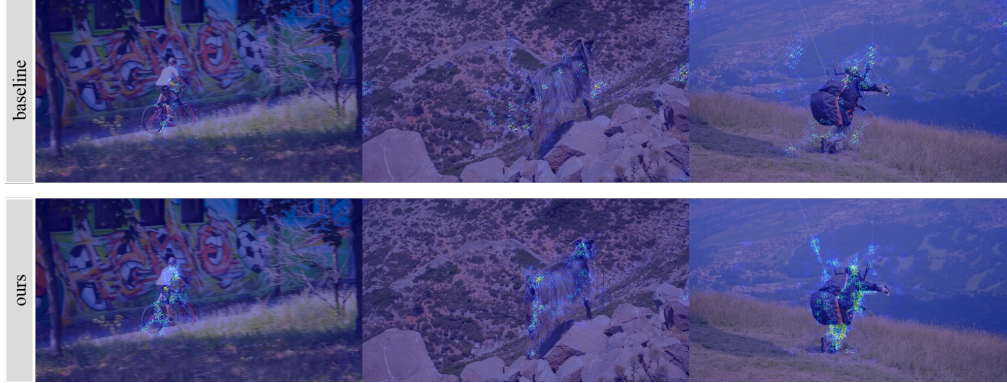

Figure 6: Cycle-ERF of frames w.r.t. the initial reference object masks in DAVIS17.

cyclic mechanism suppresses the accumulative error from problematic reference masks. From the second video, we see the cyclic model can depicts the boundary between foreground objects more precisely, which is consistent with the quantitative results. Further, our method can successfully segment some challenging small objects (caught by the left woman's hand). Readers can refer to our supplementary material for more qualitative comparison.

**Cycle-ERF analysis.** We further analyze the cycle-ERF defined as Equation (6) on different approaches. We take the initial mask as the objects to be predicted and take a random intermediate frame and an empty mask as reference. Figure 6 visualizes the cycle-ERFs of some samples. Compared with baseline, our cyclic training scheme helps the network concentrate more on the foreground objects with stronger responses. This indicates that our model learns more robust object-specific correspondence. It is also interesting to see that only a small part of the objects is crucial for reconstructing the same objects that were in the initial frames as the receptive field focuses on the outline or skeleton of the objects. This can be used to explain the greater improvement of contour accuracy using our method, and also provide cues on the extraction from reference masks.

## 5    Conclusion

This paper incorporates the cycle mechanism with semi-supervised video segmentation network to mitigate the error propagation problem in current approaches. When combined with an efficient and flexible baseline, the proposed cyclic loss and gradient correction module achieve competitive performance-runtime trade-off on two challenging benchmarks. Further explanations can be drawn from a new perspective of cycle-ERF.

## Broader Impact

As we can foresee, with the development of 5G communication, video-based media industry will develop fast in the future. The advancement of accurate and fast semi-supervised segmentation will be helpful in modern video editing software and provide real-time online segmentation solution to stream media in video live applications. Consequently the online user experience can be improved. However, there also exists the risk that video segmentation technology is utilized in the scenario of illegal shoot and malicious edit, thus the personal privacy are more likely to be exposed and tracked.

## Acknowledgement

Funding in direct support of this paper: China Major Project for New Generation of AI Grant (No.2018AAA0100400), National Natural Science Foundation of China (No. 61971277) and Adobe Gift Funding.

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
