[Supplementary Material]

# Supplementary Material

## Appendix

### A.1 Performance with different cyclic reference set

In our implementation, we set the cyclic reference set for training as $\widehat{\mathcal{X}}_t = \{X_t\}, \widehat{\mathcal{Y}}_t = \{\widehat{Y}_t\}$ for simplicity. We compare this configuration with a more complicated cyclic reference set by combining all predicted masks and its corresponding frames appearing before the $t$-th frame as the cyclic reference set. The results on DAVIS17 validation set [1] are shown in Table 1. We see there is no distinctive difference between the performance of models trained under the two schemes. The model trained with more reference in the cycle achieves higher IOU with groundtruth, while the simplified training scheme results in better contour accuracy.

### A.2 Qualitative Improvement with gradient correction

A more detailed comparison on gradient correction can be found in Figure 1. From the results, we observe that correction process can effectively suppress some false segmentation area and append segmentation of small part of objects. This observation indicates the gradient correction is beneficial to detailed segmentation.

### A.3 Mask reconstruction with cycle-ERF

We go further to investigate the cycle-ERF in this section. From Figure 6 in the main paper, we see the regions with high response intensity mainly focus on the outline of the objects in our method. In contrast, the cycle-ERF map of the baseline model is more dispersive, with some high response areas in the background. Since the cycle-ERF is obtained by the error minimization process of initial mask, this phenomenon indicates that the feature learnt by the baseline model is not robust enough thus incorrectly associate some background information with the foreground. To demonstrate this claim, we conduct an experiment by reconstructing the mask of the initial frame with the reference of selected frame and part of its cycle-ERF information.

Formally, we define the reference frame set as $\mathcal{X}_l = \{X_l\}$, and define two different reference mask set, $\mathcal{Y}_l^{ex} = \{ReLU(\widehat{Y}_l^M) \odot (1 - Y_l)\}$ and $\mathcal{Y}_l^{in} = \{ReLU(\widehat{Y}_l^M) \odot Y_l\}$, where $Y_l$ is the groundtruth mask on frame $X_l$, $\odot$ denotes elementwise production. Therefore, $\mathcal{Y}_l^{ex}$ denotes the cycle-ERF not covered by the specific objects and $\mathcal{Y}_l^{in}$ is the cycle-ERF inside the objects.

In Figure 2, we show some qualitative reconstruction results and comparison with baseline methods on DAVIS17 validation set. We observe that when only the cycle-ERF out of objects are allowed as reference, the baseline can roughly segment the whole objects while ours can only recover a small part. In contrast, when only the cycle-ERF covered by specific objects is taken into account, our method performs better than baseline. This comparison results further demonstrate that baseline method extract more background information to help segment objects, while our model trained with cycle consistency learns more accurate and robust object-to-object correspondence to deal with VOS problem.

| $\widehat{\mathcal{X}}_t$ | $\widehat{\mathcal{Y}}_t$ | $\mathcal{J}(\%)$ | $\mathcal{F}(\%)$ | $\mathcal{J}\&\mathcal{F}(\%)$ |
|---|---|---|---|---|
| $\{X_t\}$ | $\{\widehat{Y}_t\}$ | 68.7 | **74.7** | **71.7** |
| $\{X_i \mid i \in [2,t]\}$ | $\{\widehat{Y}_i \mid i \in [2,t]\}$ | **69.0** | 74.1 | 71.6 |

Table 1: Comparison between different cyclic reference set for training

Figure 1: Qualitative results of gradient correction, the right column shows the zoom up areas.

### A.4 More qualitative results and failure cases

In Figure 3, we show some additional qualitative segmentation results on DAVIS17 validation and test-dev set, showing that our method are suitable for multiple close objects and objects with fast and large motion. Figure 4 shows some failure cases of our method, although combined with cyclic loss and gradient correction, the network can not handle extremely narrow and small objects (e.g. the brassie in the man's hand in the second row), meanwhile, as shown in the first row, our method can suffer from cases where the specified objects are severely occluded by obstacles in the foreground.

## References

[1] Jordi Pont-Tuset, Federico Perazzi, Sergi Caelles, Pablo Arbeláez, Alexander Sorkine-Hornung, and Luc Van Gool. The 2017 davis challenge on video object segmentation. *arXiv:1704.00675*, 2017.

Figure 2: Visualization of initial mask reconstruction results with different settings. The specified objects are car and goat respectively.

Figure 3: Additional qualitative results on DAVIS17 validation and test-dev set

Figure 4: Failure cases of our method.