[Reviews · NeurIPS 2020]

Review 1

Summary and Contributions: This paper proposes to incorporate the cycle mechanism with semi-supervised VOS to mitigate the error propagation problem in current approaches. A gradient correction module is presented to further boost the model performance. A new visualization method, ie., cycle-ERF is developed to offer explainability on the impact of cyclic training. After reading the feedbacks and other reviewers comments, I insist my initial rating.

Strengths: The proposed cyclic loss, gradient correction module, cycle-ERF, etc., is clearly presented and easy to understand. While the overall performance is inferior to the SOTA methods, the basic idea seems to be interesting, such as cycle-ERF.

Weaknesses: 1. Please visulaize the mask update component during the gradient correction procedure´╝îie,. the second term in Equ. 5. 2. Please compare with STM[5] inTab. 1 and Tab.2.

Correctness: The mathematical derivations seem to be correct.

Clarity: The paper is well written.

Relation to Prior Work: Prior work is clearly discussed.

Reproducibility: Yes

Additional Feedback:


Review 2

Summary and Contributions: The paper proposed several improvements for video object segmentation pipelines. In particular, the main technical contributions include: 1) a cycle consistency term for limiting error propagation, 2) a gradient correction module and 3) a visualization tool which offers explainability of the prediction. **after rebuttal** My concerns(motivation, visualization) are mostly addressed in the rebuttal. I encourage the authors to improve the comparison with STM, and provide more visualization, if accepted.

Strengths: - The paper is overall well-written, the idea is clearly presented. - The proposed cycle consistency is a simple idea, while seems to be effective for video object segmentation. - The proposed visualization technique is also interesting, and might help to increase the explainability of other VOS systems, which is potentially helpful to identify the current limitations. - Besides, the code is provided, which is helpful to understand the implementation details.

Weaknesses: - The motivation of the proposed gradient correction is not clear to me. From the experiments in table 3, the module seems to be only marginally helpful. It's also not clear to me what's impact of the module on visual results, some ablated qualitative results similar with figure 5 would be helpful. - I feel the connections between three proposed components are weak. the individual improvement from each component is quite incremental. From figure 5, there are only minor differences compared to the baseline(STM). Besides, results of the baseline(STM) should be listed in Table1 and 2 to facilitate the comparison.

Correctness: Seems correct to me.

Clarity: The paper is clearly written.

Relation to Prior Work: The technical contribution is clearly stated.

Reproducibility: Yes

Additional Feedback:


Review 3

Summary and Contributions: The authors propose to use the cycle consistency to improve video object detection in the semi-supervised setting, meaning that the groundtruth mask is given for the first frame during inference. The method builds on top of the successful STM framework [5], and adds the said cycle consistency during both training and inference. Experiments are done on the benchmark Youtube VOS and DAVIS17 datasets. The authors claim novelty in: - The cycle consistency to counter the error propagation problem. - A "gradient correction module" that helps with online inference. - A cycle-ERF visualization method to explain the efficacy of the cyclic training. I most agree with the authors' assessment on the claim. However, I am slightly unconvinced by the novelty and the experimentation (see below).

Strengths: - Problem statement: video object segmentation is a well researched problem in computer vision, and is therefore relevant to NeurIPS. - Novelty: the claimed novelty in introducing cycle consistency to the semi-supervised video object detection problem seems novel and makes sense, so is the gradient correction module. I believe that the novelty, tho somewhat more application than theoretical, is sufficient. - Sufficient experimentation: The experiments were done on 2 benchmark datasets, and there are detailed ablations on the claimed improvements. It shows that both the cycle consistency and the gradient correction modules each improves over the baseline and are complementary.

Weaknesses: I have reservations on the results reported in this paper: - Worse than vanilla STM [5]: Although the paper provides sufficient experiments, the vanilla STM achieves 79.4 J&F on YoutubeVOS and 81.7 J&F on DAVIS2017. Why does this proposed "improved version" of STM perform a lot worse? According to ln 189, the authors modified the STM, but why modify it so that it performs much worse? And also, the vanilla STM does not even show in the tables in this work. Please elaborate. - Weak experiments: disregarding the vanilla STM, the method also does not seem to improve over existing methods. Sure, it can be argued that existing methods benefit from extra data. But that practice is the standard approach, whats the reason why this paper does not use extra training data? I did notice that the proposed method is faster than the references. I do like the cycle consistency idea together with the correction module. I hope that I am missing some important details here. But it does not make sense to recommend acceptance if the extension is worse than the baseline.

Correctness: No real issue. Experiments follow established practices.

Clarity: Yes, mostly. Small clarify issues here and there (see additional feedback).

Relation to Prior Work: Yes, a wide range of existing video object segmentation works have been highlighted.

Reproducibility: Yes

Additional Feedback: - ln 7: why are offline pipelines more efficient than online methods? - Figure 1: Caption? what am I seeing here? That the occluded camel is also segmented red? This could have been a semantic segmentation instead of an instance segmentation task. In the former case, the result is perfectly fine. - Figure 2 is very confusing. I took me a while to figure out where to even start reading the diagram. There are loops and bi-directional arrow everywhere. Maybe enumerate the edges and explain them in the caption? - ln 121: during training? Which training? The fine-tuning of the model on the first frame or the overall training on the training set? - In table 1 and 2, doesn't STM-cycle use online learning / fine-tuning as well? **************************************************************************** POST REBUTTAL / REVIEWER DISCUSSION COMMENTS Thanks for the explanation regarding the discrepancy in quality numbers. After reading the rebuttal and discussing with my fellow reviewers, I view this work more favorably. Should this work be accepted, I'd urge the authors to: - Still list the STM numbers in table 1 and 2. It feels a little bit unsure to base the work on STM but then leave out the STM baseline in the main tables. The authors can explain the reason of the discrepancy in the paper. - Add the experiments using extra data as well. My fellow reviewers and I entrust the authors to do so. Good luck!


Review 4

Summary and Contributions: This paper given the first frame of a video with an accurate reference mask proposes to obtain masks for the subsequent frames of the video in a semi-supervised way. The method relies on cycle consistency both in training and also during inference to further boost the accuracies. During training the ground-truth masks from a subset of frames are available, and in the inference only the first frame's mask is available. after rebuttal: I keep my rating and lean towards acceptance of this paper. If the paper is accepted, please improve the paper by incorporating the feedback from the review process.

Strengths: The paper is well organized, proposing a cycle consistency to an interesting problem. Cycle consistency is a popular approach for various applications, but to the best of knowledge, it has not been used for this problem with the proposed way. The online approach is also valuable especially for videos that will have domain gap with the training dataset. I believe it is relevant to the NeurIPS community especially for the computer vision area.

Weaknesses: The third contribution is not clear to me, cycle-ERF. First, I'm not sure how it works. The equation 5 is iteratively used on the prediction loss and YL is updated on the last iteration? Because it's a mask correction step, isn't it expected that the important regions appear around the objects. Why does the baseline model have activations far off from the object? Also it could have been better to qualitatively measure the cycle-ERF results, by finding the IoU of activations and ground-truth masks. In the experiments, is it possible to use the proposed method with extra data? Also STM results are missing, results trained by you and results provided by the paper. Results in STM paper [5] are quite good, significantly better than the ones presented in this work, I'm worried why they weren't in the Tables? If STM is significantly slow with good results, it should still be added.

Correctness: Claims are correct to the best of my knowledge. Experimental set-up and ablation studies look correct, in the exception of missing STM results.

Clarity: Yes, the paper is very easy to read.

Relation to Prior Work: This work uses cycle consistency for training and inference, and for inference it can be beneficial especially for out-of-domain videos. Unsupervised Video Interpolation Using Cycle Consistency, ICCV 2019, proposes a cyclic constrain for video interpolation with the same intuition in both training and testing, it should be added to the related work. The comparison with STM [5] is not clear. This paper also uses STM but the Table 1 and 2 do not include STM results nor the baselines trained by the authors.

Reproducibility: Yes

Additional Feedback:

[Author Response · NeurIPS 2020]



Figure 1: Qualitative results of gradient correction, the right column shows the zoom up area in left figure.

We appreciate all reviewers for the concrete and constructive comments. Following are our response for the concerns.

**[Q1](R1, R2, R3, R4)** The results reported in original paper of STM is much better than the implemented baseline and
should be included in main results.

**[A1]** We insists on our reimplemented version of STM as baseline in our paper for following reasons. (1) The original
STM did not open source training code and we find it hard to reproduce the results reported in the paper even when
the training settings and data are aligned with the original paper. (2) The vanilla STM achieves good results mainly
due to large amount of extra data (COCO, VOC, ECSSD etc.) for training, this results in unfair comparison with other
methods since the data source are not the same. Therefore, we retrain our implemented STM model solely on training
data of Youtube-VOS and DAVIS, which is the most common intersection of data source from previous works.

On the other hand, we agree that the original results in STM paper should be appended in main results for more complete
comparison. Additionally, since vanilla STM open source the inference script and pre-trained model, we can combine it
with our online gradient correction module in inference stage, we find this still brings around **1.9** boost in $\mathcal{J}\&\mathcal{F}$ score
($\mathcal{J}$ from 79.2 to **81.2** and $\mathcal{F}$ from 84.3 to **86.1**). This shows our method is general and can bring improvement for either
the vanilla STM or our reimplemented version. These results and related discussion will be appended in updated paper.

**[Q2](R1, R2)** More visualization results on gradient correction module are required.

**[A2]** We visualize examples of the qualitative effect of gradient correction in Fig 1, we can see gradient correction can
effectively suppress some false segmentation area and append segmentation of small part of objects.

**[Q3](R2)** The impact and motivation of gradient correction seems weak.

**[A3]** One core motivation of gradient correction is to mitigate the effect of low-quality reference mask during inference?.
We emphasis that gradient correction can not only bring improvement for well-trained model, but also makes the model
robust for partially low-quality reference sets. Please note the results of A.2 in Supplementary material Table 2, when
some of the reference masks in the memory are replaced by rough bounding boxes or inaccurate prediction, gradient
correction can effectively help the model yield high-quality prediction (at most $+7.3$ improvement over the perturbed
model). Besides, it should be noticed that gradient correction is also helpful for the inference of vanilla STM (see **[A1]**).

**[Q4](R2)** The connection of three contributions seems weak.

**[A4]** The three contribution follows a naturally logical relation. The offline training scheme is first proposed and then it
is extended to an complementary online version, and the cycle-ERF is derived from online update. Further, **all three**
**contributions aim at better utilization of the information from previous reference masks.**

**[Q5](R3, R4)** Why not include extra data for training as other methods?

**[A5]** We agree including extra data source for training will boost the segmentation quality (e.g. **When MSRA10k**
**is included, the performance on of our model on DAVIS17 validation set can be boosted from 71.7 to 76.2**).
Nevertheless, the problem is there is no unified protocol constraining the allowed data source in the problem of video
object segmentation for academical comparison. As a results, different method may rely on auxiliary data from different
domain or tasks. This makes it difficult to conduct fair comparison since we can not align the data setting of all previous
work. Therefore, we decide to report results of a purely trained model with DAVIS and Youtube-VOS data since they
are the commonly used and necessary data source of previous works. We will also open source our training code and
trained model, hoping this can be a more fair reference results of STM for future works.

**[Q6](R4)** The cycle-ERF is not clear.

**[A6]** The generation of cycle-ERF is nearly identical to that of gradient correction (Eq. 5) except for the cyclic reference
set is an empty mask as "zero prior" and is gradually updated to show crucial area. In some case the cycle-ERF of a
trained model highlights regions away from objects since it requires some contextual to better distinguish the objects
under such "zero prior" condition.

Finally, we will carefully polish the writing and append related reference according to the suggestion from reviewers.

[Meta-Review · NeurIPS 2020]

After rebuttal and discussion, all four reviewers have arrived to the consensus that the paper should be accepted. The reviewers acknowledge the nice idea and rationale behind the paper. Initially, the experimental results were not entirely convincing, but the extra experiments in the rebuttal showed that the improvements found were due to the novel method. This decision was verified by the AC and SAC. Based on feedback from the reviewers, the authors are requested to - make clear in the experiments the full comparison of their work versus STM and explain any discrepancies - provide more visualizations to help clarify their text and explanations. - add the recommended related work The reviewers also strongly advise that authors also add the experiments using extra data to make the comparisons more meaningful.